# Cognitive Impairment Is Associated with AMPAR Glutamatergic Dysfunction in a Mouse Model of Neuronal Methionine Synthase Deficiency

**DOI:** 10.3390/cells12091267

**Published:** 2023-04-27

**Authors:** Ziad Hassan, David Coelho, Carine Bossenmeyer-Pourié, Karim Matmat, Carole Arnold, Aurélie Savladori, Jean-Marc Alberto, Rémy Umoret, Jean-Louis Guéant, Grégory Pourié

**Affiliations:** 1Inserm UMRS 1256 NGERE–Nutrition, Genetics, and Environmental Risk Exposure, University of Lorraine, F-54000 Nancy, France; 2National Center of Inborn Errors of Metabolism, University Regional Hospital Center of Nancy, F-54000 Nancy, France

**Keywords:** methionine synthase, vitamin B12 metabolism, inherited disorders, neurologic dysfunction

## Abstract

Impairment of one-carbon metabolism during pregnancy, either due to nutritional deficiencies in B9 or B12 vitamins or caused by specific genetic defects, is often associated with neurological defects, including cognitive dysfunction that persists even after vitamin supplementation. Animal nutritional models do not allow for conclusions regarding the specific brain mechanisms that may be modulated by systemic compensations. Using the Cre-lox system associated to the neuronal promoter Thy1.2, a knock-out model for the methionine synthase specifically in the brain was generated. Our results on the neurobehavioral development of offspring show that the absence of methionine synthase did not lead to growth retardation, despite an effective reduction of both its expression and the methylation status in brain tissues. Behaviors were differently affected according to their functional outcome. Only temporary retardations were recorded in the acquisition of vegetative functions during the suckling period, compared to a dramatic reduction in cognitive performance after weaning. Investigation of the glutamatergic synapses in cognitive areas showed a reduction of AMPA receptors phosphorylation and clustering, indicating an epigenomic effect of the neuronal deficiency of methionine synthase on the reduction of glutamatergic synapses excitability. Altogether, our data indicate that cognitive impairment associated with methionine synthase deficiency may not only result from neurodevelopmental abnormalities, but may also be the consequence of alterations in functional plasticity of the brain.

## 1. Introduction

One-carbon metabolism nutritional and inherited disorders with reduced availability in methyl donors and methylation capacity of DNA and other substrates during gestation are known to induce neurodevelopmental abnormalities such as *spina bifida* and brain altered morphological and functional development [1,2,3,4]. For several decades, this led to the supplementation of pregnant women or food fortification in so-called nutritional “methyl donors” such as folate (vitamin B9) and vitamin B12 in different countries [5,6]. From a scientific point of view, the underlying mechanisms highlighted the major implication of the metabolism of vitamins B9 and B12 in developmental gene expression through epigenetic regulation [6,7,8,9]. In addition, brain dysfunction could have at least two origins, a typical developmental defect in which specific developmental genes are implicated [10,11], and/or synaptic dysfunctions in which the neurodevelopment runs well [12]. It is well established that nutritional and inherited disorders of folate and vitamin B12 metabolism produce neurological and cognitive disorders through impaired methionine synthase activity [9,13]. In contrast, the effect of supplementations or fortifications on cognitive improvement is a matter of debate [14,15,16].

In addition to gestational models, more specific approaches are needed to investigate unidirectional deficits rather than global dysregulation. A nutritional vitamin deficiency affecting various organs such as the gut, the heart or the brain could partially be compensated/modulated by circulating actors and/or liver biochemical adaptation [17,18]. Additionally, specific inherited disorders that affect the intracellular metabolism of vitamin B12 are described, such as cblC and cblG genetic defects [3,19]. These pathologies lead to hematologic and neurologic diseases, and contrary to metabolic problems, neuronal functions do not respond to classical treatments based on vitamin injections. Thus, a more specific model than nutritional or gestational models of one-carbon metabolism defect could help investigate specific brain persistent dysfunctions. Nevertheless, systemic knock-out models of enzymes involved in the methionine cycle often lead to embryonic lethality, suggesting that a mouse model of tissue-specific gene knock-out may be more relevant to study the role of methionine synthase in neuronal function.

Methionine synthase is a key enzyme of the one-carbon metabolism, linking folate and methionine cycles and using vitamin B12 as a co-factor. It catalyzes the transformation of homocysteine in methionine using a methyl group from methyltetrahydrofolate. Knocking-out of this enzyme in one organ/tissue allows studying the specific consequences of its impaired expression, in contrast to usual approaches with global nutritional or genetic models [9,20]. Neurologically, cognitive outcomes are predominant manifestations of inherited errors of cobalamin metabolism (IECM) with impaired methionine synthase activity, such as cblG and cblC defects. The mechanisms underlying the age-dependent high variability of neurological and cognitive manifestations, and the limited responses to conventional therapy of IECM are poorly understood. Even if the dysregulation of several metabolic actors in the plasma or the liver could be corrected in nutritional models after a switch to a normal diet, neurological outcomes such as cognitive disorders are not corrected in both nutritional rodent models and human inherited disorders [21,22,23].

Cognitive functions are mainly supported by glutamate circuits, especially in the limbic system, including the hippocampus formation linked to the frontal cortex. Among the diseases related to one-carbon metabolism, numerous studies highlighted a so-called fetal programming effect leading to abnormal brain circuits during gestation or at birth [7,11,21]. Epigenetic mechanisms based on a lower methylation status through methionine and S-adenosine-methionine down-regulation could lead to modified gene expression during gestation. Thus, according to permissive windows of time, a correction through structural/neuroanatomic plastic reactions could occur in favor of a diet modification or vitamin supplementation. However, later in age, or without any permissive correction/adaptation, numerous functional defects could persist or be revealed during adult life and/or aging [24,25,26]. These neurodegenerative perturbations could be as different as neuro-psychiatric retardation, cognitive impairments, accelerated aging decline, and Alzheimer’s disease [27]. In addition, genetic mutations related to actors of the cobalamin metabolism, especially cblC and cblG defects, also lead to dramatic brain dysfunctions and death in infants or young adults despite cobalamin treatments [22].

Compared to global approaches with nutritional models, a brain-specific model of dysregulation of the methionine cycle could focus on neurologic mechanisms leading to functional defects. They may be related to neurodevelopmental/structural damages during the brain’s construction, or rather be due to functional dysregulations in synaptic plasticity, or both. A cellular/molecular mechanistic approach could propose putative therapeutic targets since, to date, no efficient correction of brain dysfunction is available under the dysregulation of the methionine cycle. In this context, we aimed to explore synaptic and plasticity dysfunctions as a putative corpus of mechanisms that may explain the persistent neuro-functional impairments related to the expression and activity of MS in a brain-specific KO mouse model. Using the Cre depletion of two exons of the Mtr gene, under the control of the Thy 1.2 promoter, allows us to focus on the role of MS on neuronal mechanisms underlying brain outcomes. Offspring born using this model allowed constituting two groups of developing mice, one depleted for the expression of the methionine synthase (MS) in neurons (named KO-Mtr, or KO) and a second with a full expression of this enzyme (named WT).

## 2. Materials and Methods

### 2.1. Animals and Tissue Collection

Experimental mice were obtained from males and females C57BL/6 with the Cre/Lox system using *Thy-1.2* as a promoter of the CRE recombinase, in order to generate *Mtr*-knock-out (*Mtr*-KO) in brain tissue. Wild type animals (WT) were obtained as littermates of KO, when they did not receive and produce the Cre recombinase (Figure 1A). Genitor females and males were obtained from ICS (Institut Clinique de la Souris, Strasbourg, France). Females carried the “floxed MTR-exons” and males carried the Cre recombinase under the *Thy-1.2* promotor. Crossing these females and males gave a theoretical range of 50/50% WT and KO offspring. Experiments were performed on offspring during postnatal growth from 5 days of age (D5) and until 37 days of age (D37). Mice were maintained under standard laboratory conditions, with a normal 12-h light/dark cycle, food and water *ad libitum*. Neonates were kept with mothers during the lactation period and weaning time was considered at D21 after birth. Both male and female offspring were considered for experiments, and when data revealed no sex difference, they were pooled. Animals were treated in accordance with the National Institute of Health Guide for the Care and Use of Laboratory Animals, in an accredited establishment (Institut National de la Santé et de la Recherche Médicale, U1256) according to French governmental decree 2013-118 and authorization number *APAFIS*#2776-2015111915482808. Mice were euthanized at D37 with an overdose of isoflurane and decapitated in order to collect tissues. Brains were rapidly harvested and placed on ice. In the beginning of the study, whole brains were considered to measure total weights (Figure 1E), relative amounts of *Mtr* gene mRNA (Figure 1F), and methionine synthase protein (Figure 1G). After these global brain investigations, the two hemispheres were separated. Left hemispheres were frozen in methylbutane and stored at −80 °C for further cryo-sections. Right hemispheres were microdissected on ice, and brain sub-areas such as the hippocampus and the cerebral frontal cortex (the whole cortex was dissected, cut in two according the parietal zone, the frontal part was considered) were immediately frozen in liquid nitrogen and stored at −80 °C until biochemical/molecular analyses (see figure legends for precision on sub-areas use in each experiment). Blood samples (200 µL) were collected in EDTA tubes from the sub-mandibular vein before decapitation.

### 2.2. RNA Analysis and Quantitative RT-PCR Analysis

Total RNA (500 ng) was purified from nitrogen frozen brain tissues from microdissections of WT and KO mice (*n* = 5 for each group) with the iTaq™ kit SYBR^®^ Green One-Step (Takara, Kusatsu, Japan). Specific amplifications were performed using the following primers ordered from Bio-Rad (Hercules, CA, USA): Mtr primers (exon 4-6 Forward ACACTTGGCCTACCGGATG and Reverse CCAGCCACAAACCTCTTGAC; exon 29-30 Forward CTGAGCCCATAGCCACCTTC and Reverse CCAAAGCAGGCAACAGCAAA). Cycle threshold (Ct) was determined for each sample and the expression of the genes of interest was normalized to Pol2 gene (Forward AGCAAGCGGTTCCAGAGAAG and Reverse TCCCGAACACTGACATATCTCA) using the 2^−ΔΔCt^ method. Results were expressed as arbitrary units (AU) by calculating the ratio of crossing points of amplification curves of mRNA and internal standard.

### 2.3. Protein Extraction and Wes Protein Analyses

Nitrogen-frozen brain samples (microdissections) were solubilized in RIPA lysis buffer complemented by phenylmethanesulfonyl fluoride 1% (Sigma-Aldrich, Saint-Quentin-Fallavier, France), Sodium orthovanadate 1% (Sigma) and Phosphatase Inhibitor Cocktail 0.5% (Roche, Mannheim, Germany). The protein concentration was determined using BCA Protein Assay kit (Interchim, Montluçon, France). Considering the weak amount of total protein from small samples such as the hippocampus, analysis was performed using Wes Simple Protein system (Bio-Techne, Minneapolis, MN USA). The Wes automated capillary-based size sorting system was used to analyze protein expression. 0.4 ug/uL of total protein was loaded to WES 25-well plates for separation following provider instructions. Primary antibodies were diluted as mentioned: Methionine synthase (ThermoFisher, Illkirch, France; 25896-1-AP, 1:100); AmpaR1 (Abcam, ab31232, 1:100); AmpaR2 (Abcam, ab133477, 1:700); phospho-AmpaR1 ser845 (PhosphoSolutions, P1160-845, 1:100); phospho-AmpaR2 tyr876 (ThermoFisher, Illkirch, France; Pa5-17096, 1:10); post-synaptic density-95 (PSD-95) (ab13552, mouse monoclonal, 1/700, Abcam). The total amount of proteins per lane was normalized using alpha tubuline (Cell Signaling, Ozyme, Saint-Cyr-L′Ecole, France; #2144S, 1:100). HRP labeled anti-rabbit or anti-mouse (ProteinSimple, Bio-Techne, Minneapolis, MN USA) secondary antibodies were used according to manufacturer recommendations. The relative amount of each protein was analyzed through the areas under peaks from the chemiluminescence chromatograms by the Compass for SW software v6.0.0 (ProteinSimple, USA). Results are presented as a virtual blot generated by Compass for SW software.

Some antibodies were revealed inefficient using the Wes technique; therefore, a classical Western blot analysis was performed for NMDAR1/R2 and CREB/p-CREB. Western blot analyses were performed on nitrogen frozen isolated brain structures. Tissue was solubilized in RadioImmunoPrecipitation Assay (RIPA) Lysis buffer containing 140 mM NaCl, 0.5% (*w*/*v*) sodium deoxycholate, 1% (*v*/*v*) Nonidet P-40, 0.1% (*w*/*v*) SDS, and protease inhibitors (Complete, Roche Applied Science, Meylan, France). After homogenization, samples were lysed by three cycles of freezing/thawing and finally centrifuged at 4 °C for 30 min at 15,000× *g*. The protein concentration in the supernatant was determined using the BCA protein assay kit (Pierce, Interchim, Montluçon, France). Moreover, 40 µg protein samples were mixed with an equal volume of 2× Laemmli buffer, denatured by heating the mixture for 5 min at 100 °C, and then resolved by 12% SDS-PAGE. The separated proteins were transferred using a Mini Trans-Blot cell onto polyvinylidene fluoride membrane (Immobilon-P, Millipore, Merck, Fontenay-sous-Bois, France), and the membranes were blocked for 1 h with Tris-buffered saline (pH 7.4) and 0.1% (*v*/*v*) Tween 20 (Tris-Buffered Saline Tween TBST buffer) containing 5% (*w*/*v*) bovine serum albumin. The polyvinylidene fluoride membranes were then incubated overnight at 4 °C with a primary antibody against one of the following proteins: N-methyl D-aspartate receptor 1 (NMDAR1, Pa5-34599, rabbit polyclonal, 1/1000, Invitrogen Thermo Fisher Scientific, Waltham, MA, USA); phospho-NMDAR1 (ABN99, rabbit polyclonal, 1/50, Invitrogen Thermo Fisher Scientific, Waltham, MA, USA); NMDAR2 (AB1592P, rabbit polyclonal, 1/1000, Chemicon); phospho-NMDAR2 (AB5403, tyr1472-rabbit polyclonal, 1/50, Chemicon); CREB (Cell Signaling #4820, 1:100); phospho-CREB ser133 (Cell Signaling #9198, 1:50), alpha tubuline (Cell Signaling #2144S, 1:100) was used as an internal standard. Polyvinylidene difluoride membranes were incubated for 1 h at room temperature with the corresponding horseradish peroxidase-conjugated pre-adsorbed secondary antibody (1/5000, Molecular Probes, Eugene, OR, USA). Quantity One software, associated with the VersaDoc imaging system (Model 1000, Bio-Rad Laboratories, Hercules, CA, USA), was used to quantify the signals.

### 2.4. Behavioral Tests

Various behavioral tests were used starting from first significant movements of neonates at D5 and until D36 after weaning (D21). All tests were performed between 08:00 and 11:00 h. a.m., which corresponds to the beginning of the animals facility light period (end of animal active period before sleeping periods occurring after 12:00 a.m.). Standardized tests adapted for specific windows of developmental times were used [21,28,29,30], with *n* = 9 Wild-type and 8 *Mtr*-KO. Except for the evaluation of young mice coordination, all behavioral tests were recorded using a video-tracking system (Viewpoint) allowing a high standardization between runs.

Coordination tests in young ages: The static righting reflex was studied as described by Blaise et al. [21]. Briefly, the time needed by a young mouse to right itself from a position on its back to a supine position was recorded at postnatal days 5, 7, and 9. The negative geotaxis was also tested from postnatal days 5, 7, and 9, with a 30 min interval from the righting reflex test. Each mouse was positioned with the head downward on an inclined plane with a 20% slope. The time needed for the young mouse to turn completely and reach a position with the head upward on the plane was measured. The duration of these tests was limited to 120 s (cut-off).

Muscular abilities with the Suspension test: This test was performed at D16, D18, D20, during the last half of the sucking period when young mice exhibit the highest activity in the home cage. Each mouse was proposed to grip with the front paws a horizontal rod (1 mm in diameter) fixed at 60 cm above the table. A foam was disposed under the rod to avoid dangerous falls. The time the mice spent in suspension with their front paws on the rod was recorded.

Open-field test: This test is performed at 21 days of age (D21) to assess the amount of locomotion and the level of motivation to explore in rodents. It is a circular area boarded by grey-plastic walls (35 cm high). On the floor of the area are designed 4 quarters drawing 8 zones (4 near the walls and 4 in the center). Individuals are gently dropped in the middle of the open field and followed by the video-tracking system for 5 min. The location (protected near walls versus unprotected in the middle) and the number of zones visited (entries, time spent, distance moved) were recorded. Eight zones were considered on the floor, four in the center (“unprotected zones”) and four in the periphery (“protected zones”), to investigate anxiety-like cues. The apparatus is cleaned with 30% alcohol between each individual.

Elevated plus maze: It consists of a maze made with two corridors (9 cm large and 1 m long) crossing each other, designed with 4 arms opposed two by two. Two arms are open without walls, and the two others are closed by dark-grey plastic walls (35 cm high). This maze is elevated 80 cm above the floor. 22-day-old (D22) individuals were placed in the middle of the crossing corridors facing an open arm and allowed to explore the maze for 5 min. Its behavior was recorded with a video-tracking system that allows optimum standardization. This test gives some cues reflecting the basal stress level of individuals and information concerning locomotion (in “protected” zones with walls versus in “unprotected” zones without walls, and in the center of the apparatus named “intermediate”) and movement coordination in case of falls. The apparatus is cleaned with 30% alcohol between each individual.

Learning performances: were evaluated using two behavioral tests specifically adapted to postnatal development time windows. Firstly, the Homing Test was performed during the suckling period of young mice, immediately after the beginning of the walking behavior (11-day-old, D11), and at the end of the suckling period (19-day-old, D19). The apparatus is made of a plastic corridor (6 cm large, 80 cm, walls 12 cm high) virtually divided into three equal zones. At each end of this corridor, a small compartment contains the litter used for housing cages, fresh or used, in the home cage of tested mice. Using transversal plastic walls with holes, putative contacts between pups and cage litters are not allowed. Comparable amounts of litter are randomly placed at the right or left side of the corridor and each mouse is placed firstly in the central zone for 5 min. Different parameters, such as the time spent in the different zones, attest to the acquisition and learning of the homing behavior based on the olfaction of growing pups. The apparatus is cleaned with 30% alcohol between each individual. Secondly, to confirm the learning and memory performance investigated during the suckling period, a Water Maze was used after the weaning period between 28- and 32-days-old. The apparatus consists of a square pool filled with water (5 cm deep, maintained at 25 °C). Grey plastic walls 30 cm high is used to delimit twenty-five square zones equal in size (15 × 15 cm). Open doors in walls allow communications between zones designing an ideal route from a starting zone to an exit one with additional lateral error zones. Each mouse was allowed to run the maze twice daily for five consecutive days (sessions S1 to S5 with a cut-off time of 120 s). Different parameters, such as the time necessary to escape the maze (Escape Latency), the number of errors (no-end zone entries or number of turn-around) and the number of immobilizations, attest to an efficient learning mechanism if a decrease is recorded according to the successive sessions. The cut-off time was attributed if no exit was recorded (120 s). Two days before session one, each mouse was allowed to know the apparatus in two habituation sessions (not recorded).

Depressive-like sensitivity with the Forced swim test: At 36 days of age (D36) an evaluation of mice psychic sensitivity exposed to a no-escape environmental paradigm was performed. The forced swim test or Porsolt test is composed of a circular tank (20 cm in diameter, 35 cm high) filled with water to a depth of 25 cm and maintained at 25 °C. Each mouse was gently dropped in water and allowed to swim for 5 min. No case of swim difficulty was recorded, and no mice drowned. The time of the first swim stop and the cumulative immobility were recorded.

### 2.5. LC-MS/MS Analyses

To quantify one-carbon metabolism markers, a piece of frozen hippocampus was mixed, weighted, and solubilized mechanically in 1X PBS using pellet pestel motor (Kontes), following by ultrasonic bath for 20 min at 4 °C and then centrifuge at 12,000× *g* for 20 min. The supernatant was collected for LC-MSMS analysis. Briefly, the supernatants were mixed with DTT containing internal standards, precipitated by adding cold methanol, and maintained on ice for 30 min. After decanting, the liquid phase was diluted with four volumes of 0.1% formic acid. Experiments were carried out using Shimadzu LCMS 8045 ESI Triple quadrupoles and analyzed using Insight software v3.1 (Shimadzu, Kyoto, Japan; french supplier Marne-la-Vallée, France).

### 2.6. Tissue Measurments and Immunohistochemistry

Twelve-µm sagittal cryo-sections were generated, starting from the zero plane that bisects the brain mid-sagittally. Brain structures were identified according to the Paxinos and Watson atlas for slide standardization according to the same coordinates.

For subsequent staining and labeling counts or measurements, selected slides were coded prior to analysis and the codes were not broken until experiments were completed (blind procedure). Frontal cortex cell densities and hippocampus (CA1 Ammon’s Horn and Dentate Gyrus) layer thicknesses were measured stereologically after thionin staining (one slide per animal). Images were collected under 20× magnification with an Olympus microscope connected to a digitally calibrated camera and the Cell^F^ software. For thickness measurements of CA1 and DG, the entire structures were considered and the value for each animal was the mean of 10 measures (2 slides per individual considered with the same atlas coordinate at ±100 µm). For cortical cell density, all layers under the scalp were evaluated counting all nuclei on a 500 × 500 µm square. In case of no difference between groups, the density for layer II was presented (one image on each slide; two slides per individuals, according to the atlas coordinate).

For imunohistochemistry, tissue sections were incubated in 0.1% triton ×100 in phosphate-buffered saline (PBS) for 20 min at room temperature. Slides were dipped in PBS for 10 min, then in PBS containing 10% bovine serum for 1 h, and were incubated two days at 4 °C with primary antibodies (see the Wes protein analysis section above), followed by the second-step antibodies adapted to the host of the first antibodies for 1 h at room temperature (IgG conjugated to Alexa Fluor, 1/1000; Molecular Probes) and finally counterstained with DAPI. Images were obtained using a Nikon instruments confocal microscope, Champigny-sur-Marne, France, at ×60 magnification.

### 2.7. Protein Interactions

The “Proximity Ligation Assay” PLA (Duolink^®^ in situ PLA^TM^ reagents, Olink Bioscience, Eurogentec, Angers, France) was used to visualize and quantify in situ protein interactions, following the manufacturer’s recommendations. Briefly, after fixation of the tissue section in 4% paraformaldehyde for 10 min followed by 3 washes in PBS, permeabilization was performed in PBS containing 0.1% triton for 10 min. The slides were incubated in a blocking solution for 1 h and at 4 °C for 48 h with the primary antibody diluted 1/200. A pair of oligonucleotide labeled secondary antibodies (PLA probes) was then added to the slides for 1 h at 37 °C in a humidified chamber, the ligase was then added for an extra 30 min at 37 °C, before the amplification step, 2 h at 37 °C. Cell nuclei were stained with DAPI. The PLA probes generate a signal only when the two probes are bound in close proximity. The signal from each detected pair is visualized as an individual fluorescent dot. The PLA signals can be counted and assigned to a specific subcellular location based on microscopy images (first preview with an Olympus BX51WI microscope and finally for analysis with an Nikon instruments confocal microscope, Champigny-sur-Marne, France, completed with BlobFinder freeware from the Centre for Image Analysis, Uppsala University, Sweden). As all brain tissue investigations (see above), stereotaxic atlas from Paxinos & Watson was used to control the anatomic localization of slides counted (at least *n* = 4 WT and 4 KO). As recommended, four negative controls were prepared for each experiment as follows: (1) primary antibodies only, (2) PLA probes only, (3) one primary antibody + PLA probes, (4) the other primary antibody + PLA probes. These control combinations must not show any dots to validate the experiment.

### 2.8. Statistical Analysis

Data are presented as means ± s.d. (means ± s.e.m. for behaviors for graphical clarity), and were analyzed with Statview 5 software for Windows (SAS Institute, Berkley, CA, USA). They were compared using two-way or one-way variance analysis (ANOVA) with Fisher’s test. *p* value < 0.05 was considered as significant.

## 3. Results

### 3.1. Physiological Characterization of the Mouse Model

We used a small piece of tail to genotype the pups several days after birth (i.e., usually four days) to avoid mice disturbance. This PCR experiment revealed the presence or absence of the CRE enzyme attesting to the genotype of each pup (i.e., Wild Type without CRE and Knock Out with CRE, Figure 1A,B). The first genotyping of pups with PCR was compared and/or confirmed using a protein analysis performed by the WES technique on brain extracts at D35 (example of WES protein analysis, Figure 1C). An RT-q-PCR (Figure 1F; Table 1) and a protein analysis (Figure 1G; Table 1) run for the expression of the *Mtr* gene with brain extracts at the end of the protocol (37-day-old) confirmed the initial genotype. These experiments revealed that KO mice presented significantly less mRNA, with a three-fold decrease in mean, and almost no protein of the methionine synthase (MS) enzyme than WT control mice. In parallel with these DNA, mRNA, and protein validations, a follow-up of the growing pups showed no difference in body and total brain weights between WT and KO mice (Figure 1D,E). A detailed analysis of different brain sub-structures showed that the depletion of the MS enzyme appeared not equal in all sub-regions. The cortex and hippocampus appeared to show the lowest amount of MS protein in the brains of KO mice (Figure 1G). From a biochemical point of view, some of the major markers related to the one-carbon metabolism and MS function confirmed, in the Mtr-KO model, a dramatic reduction of metabolic markers of the remethylation pathway in parallel of the reduced MS expression in the hippocampus of KO compared WT mice (Figure 1H; Table 1).

### 3.2. Behavioral Characterization

Different categories of brain functions were scanned as earlier as the detectable movements of pups in the nest starting at 5-day-old (D5). A battery of already published tests allowed us to investigate various neurological outcomes from young ages to pre-adult stages. Using the vegetative reflexes of very young pups from D5 to D9 and muscular and coordination abilities from D16 to D20 during the lactation period, the results showed that KO pups presented punctual behavioral deficits compared to WT. In each reflex test (righting reflex, Negative geotaxis), all pups presented a correct acquisition of the tested functions revealed by the decreased time necessary to execute the test from D5 to D9. However, in these two first tests, the performances of KO pups were reduced compared to WT at D5 and/or D7, with a significantly longer time to execute the test, attesting of transient retardation in the acquisition (Figure 2A,B; Table 2). The Suspension test evaluated between D16 to D20 confirmed the retardation of KO pups, since they remained a significantly shorter time suspended to the rod than WT (with a 40% reduction at D18, Figure 2D; Table 2). Nevertheless, at the end of the lactation period (D20) both KO and WT pups presented comparable muscular and coordination abilities, one more time attesting of transient retardation of neuromuscular function in KO pups.

After weaning, mice were tested for their locomotion and exploration behaviors. Both the Open-field (OF) and the Elevated-Plus-Maze (EPM) tests showed no major differences in the parameters of locomotion and the visitation of protected or unprotected zones, indicating no disturbance in the related neuronal functions following the Mtr KO. A 50% decrease in the visits of EPM unprotected zones in KO versus WT was measured, but was not confirmed with results obtained in the open field test. At this age, young mice spent a majority of time in their intermediate (central) zone of the EPM. (Figure 2E,F).

Switching to cognitive functions, two different tests revealed that learning and memory capacities of KO mice were affected. The Homing test investigates the first olfactive-dependent memory of growing pups during the lactation period. This test showed that the nest odor was less recognized and memorized in KO than in WT mice at D11, regarding visit frequency (10% vs. 35%, respectively) and the number of choices of the “home zone” (1.5 vs. 2.5 fold, respectively). Finally, the time spent by KO pups in the “home zone” was reduced compared to WT, with a significant 4-fold reduction at D11 (Figure 2C; Table 2).

The Water-maze test performed from D33 to D37 also revealed that KO mice presented poorer learning and memory performances than WT controls. Indeed, KO mice showed significantly higher escape latencies from session 2 to session 5 to run and escape the maze compared to WT (over 60 s for KO vs. decreasing from 60 to 20 s for WT, Figure 3A; Table 3). The number of errors committed appeared not significantly different, but this parameter needs to be carefully analyzed in the light of cognitive strategy. Paying attention to the traces of mice (examples of one mouse trace for each group on Figure 3B–D), it appeared that KO mice need more stops and immobilizations, especially at corners of the ideal route (Figure 3C, green + black colors) compared to WT (Figure 3D). KO mice presented 20 stops versus 10 in mean for WT in session 5, where the learning should be acquired. Such high reduction of navigation indicates a cerebral hesitation for choices, and low cognitive performances also revealed by the difference between session 1 (all mice naive) and session 5, which is characteristic of the acquisition of the hippocampus visuo-spatial learning (Figure 3B–E).

Depressive-like sensitivity was investigated using the Porsolt test. It reveals rodents’ sensitivity to developing behavioral depressive markers, especially by measuring the ability to resist resignation during a no-escape situation. The results showed that 67% of KO mice presented a high sensitivity to depressive-like status. In comparison, only 25% of WT mice did. The table summarizes the behavioral markers for psychiatric outcomes and shows the highly significant differences between highly sensitive and low sensitive mice. Taken together, these quantifications indicate that KO mice are predisposed to develop depressive-like symptoms compared to WT (Figure 3F).

### 3.3. Status of Cognitive Brain Sub-Regions

Histologically, the tissue itself appeared not dramatically affected by the MS depletion. Actually, among all brain sub-regions investigated, we only measured a slight reduction of the CA1 zone of the hippocampus, while other zones were not different (Figure 4).

As glutamate receptors are mainly implicated in cognitive functions and in numerous neurodegenerative diseases, we investigated whether they were affected in our model. In hippocampus micro-dissection, results showed that the expression of NMDA, AMPA, and PSD-95 were not affected by Mtr deletion (Figure 5A–C). In contrast, we obtained a significant reduction in tyrosine 876 and serine 845 phosphorylated AMPA receptors 1 and 2 in KO compared to WT (Figure 5D; Table 4). The evaluation of a downstream marker of plasticity related to glutamatergic synapses showed a reduced expression of the transcription factor CREB and phosphorylated-CREB in KO mice (Figure 5E,F; Table 4).

The histological investigation of the CA1 hippocampus layer showed that despite an equivalent expression level, AMPA and NMDA receptors seemed to be more clustered in WT cells and more diffused in the cells of KO hippocampus (Figure 6). Indeed, using the Duo-link technique to investigate the clustering of glutamate receptors and the PSD-95 (Figure 7), the results clearly showed a disorganization of these functional actors of glutamate synapses in the CA1 layer of hippocampus of KO mice compared to WT (Figure 8).

## 4. Discussion

Dysregulation of the one-carbon metabolism during embryogenesis or early life has been described as a major factor increasing the prevalence of neurodevelopmental abnomalities, especially regarding cognitive performances in infants and young adults [31]. Thus, in many countries, this led to supplementing pregnant women in folate or global food fortification in methyl donors [32]. In parallel, B-vitamins were designed as major factors for the one-carbon metabolism health in malnutrition and genetically inherited disorders. Indeed, impairment of vitamin B9 and B12 metabolisms could provoke epigenetic and/or epigenomic problems, often leading to neurologic dysfunctions and death [33,34,35]. In this context, it appears of great interest to distinguish between the various mechanisms and cellular/molecular actors implicated, since almost no treatment is available, except for the biochemical correction of vitamin status, which is relevant but neurologically partially inefficient [23,36].

Nutritional models depleted in vitamins give a systemic approach. The global biochemical cross-talks between organs such as the liver and brain [37,38] lead to confusing conclusions on specific brain dysfunctions [39]. In our model of neuronal Mtr KO, the protein expression of MS was dramatically depleted and almost not detectable in KO mice. Concerning littermate WT, our results showed variability in MS expression levels according to (i) individuals and (ii) brain sub-regions, with the smallest amount of protein in the hippocampus. Such variability could be discussed in light of alternative splicing and the difference in enzyme activity between individuals [40]. Despite the absence of MS expression in the brain of developing mice in the KO group, we did not measure growing differences regarding body and brain weights between KO and WT. This constitutes a major difference with nutritional models of the one-carbon metabolism dysregulation, in which developmental and/or growing differences lead to fetal programming consequences [11,21].

Since no major physical difference was observed between KO and WT mice, the behavioral characterization monitored at specific days would attest to the importance of neurofunctional putative deficits without the influence of systemic dysregulations often proposed in nutritional models. From D5 to D20, all specific behavioral tests showed that vegetative or “basic physiological” functions such as movement coordination, locomotion, and muscular capacities were only slightly disturbed in KO compared to WT. Additionally, just after weaning, the open-field and elevated-plus-maze tests revealed no major differences in new environment exploration and stress and anxiety-like behaviors. Indeed, only punctual retardations in acquiring neurobehavioral-specific functions were observed. Young KO mice presented several deficits compared to WT, and often in the early ages of each window of time tested. Instead of a dramatic deterioration of brain functions, this rather suggests a delay in the circuit’s maturation since, in each test, the performances of KO mice reached those of WT at older ages in each window of time, test by test. In the context of dysregulation of the one-carbon metabolism during gestation, several studies highlighted neurodevelopmental deficits [29,41] that could be corrected thereafter, most of the time [26]. These data suggest a transient phenomenon that could be adjusted in growing individuals. Nevertheless, among all behaviors tested, cognitive functions appeared the most affected. The visuospatial maze test indicated that KO mice of 28–32 days of age presented a dramatic reduction of their learning and memory performances to efficiently exit the maze compared to WT controls, confirming our previously published results [42]. Moreover, focusing on the route traces, it appears that KO mice markedly needed stops and spins around at cross-points of the maze, probably indicating difficulties for memory formation. In investigating the psychiatric status of mice, the forced-swim test clearly showed that KO mice presented a higher sensitivity to depressive-like syndromes than WT. This suggests a higher fragility to environmental pressure and a cognitive deficit in thought and analysis abilities [43,44]. Previous studies have shown that such cognitive deficits are often persistent and poorly corrected by classical treatments in the context of impairment of cobalamin metabolism or low methylation status [23,36]. However, we have already demonstrated that this cognitive impairment could be treated pharmacologically using this transgenic mouse model treated with the SIRT1 activating compound SRT1704 [42].

The hippocampus formation and the frontal cortex are the major brain sub-structures implicated in cognitive functions. Indeed, in our model, the hippocampus of KO mice presented a significant reduction in methionine, S-adenosine-methionine, S-adenosine-homocysteine, and an increase in homocysteine, four major actors of the methionine cycle implicated in epigenetic and epigenomic mechanisms. A slight reduction of the CA1 layer was measured in KO mice. However, a thin difference could be discussed from a functional point of view rather than related to a strong developmental deficit (see below, the Hebbian mechanism).

According to the significant reduction of cognitive performances in sub-adults, we investigated glutamatergic receptors known to drive cognition. In contrast to nutritional models, KO and WT mice displayed no difference in the expression level of AMPA and NMDA receptors [45]. Nevertheless, the phosphorylation of AMPA receptors was significantly reduced in KO mice, suggesting a dysregulation of AMPAR functionality. This proposed mechanistic is reinforced by the fact that the measurements concerning glutamate synapses in the hippocampus were performed on tissue extracts collected only three days after the maze test and immediately after the forced-swim test. The glutamatergic plasticity is based on the cross-talk between the synaptic potential driven by AMPAR and the consecutive activation of NMDAR, both active in postsynaptic density clusters (PSD) [46]. As proposed in other studies, the dysregulation of the methionine cycle through a depletion of the methionine synthase could provoke a reduction of the functionality of AMPA receptors in synaptic buttons. Then, this rather increases the level of non-functional extrasynaptic AMPAR, leading to a lowering of the synaptic potential associated to long-term depression (LTD) rather than long-term potentiation (LTP) [47,48]. These findings led us to investigate the putative difference between synaptic and extrasynaptic AMPAR in our model. Immunohistochemical labeling showed that AMPAR and NMDAR were distributed differently in tissue slides, with a better clustering with PSD-95 in WT compared to diffuse labeling in KO mice, indicating a recruitment of receptors in the global PSD known to be related to cognitive functions [49]. In this mechanistic context, it has been shown that the phosphorylation of AMPAR at specific sites promotes its functionality. On the one hand, the Src phosphorylation of AMPAR tyrosine 876 allows the receptors exocytosis from endosomes to the postsynaptic membrane with GRIP protein [50]. On the other hand, the PKA phosphorylation on serine 845 increases the AMPAR lateral translocation from extrasynaptic to active-synaptic button and PSD-95 clustering using the stargazin (TARP) protein [51]. The regulation of these various trafficking pathways has emerged as a key mechanism for activity-dependent plasticity of synaptic transmission, implicated in cognitive functions such as learning and memory [52,53]. Thus, different kinases play a central role in the dynamic of synaptic plasticity. We have previously shown that the dysregulation of the one-carbon metabolism leads to reduce the expression and/or the activity of various kinases or phosphatases, such as Erk1/2 MAPK, Src, JNK, p38 MAPK, PKA, and PPA2, through epigenomic mechanisms associated with reduced activations of some receptors, such as ERα or GR and specific micro-RNA over-expression [10,30,54,55].

In our neuronal-specific model of methionine cycle dysregulation, it appears that the neurofunctional deficits, such as cognitive dysfunctions also monitored in patients, correlate with a loss of AMPAR recruitment to the membrane. Such an AMPAR-mediated reduced synaptic activity is known to drive NMDAR-related plasticity leading to cognitive functions. In addition, this mechanistic link starting with AMPAR could also explain the slight decrease we obtained in the CA1 hippocampus thickness in KO mice, contrary to most nutritional models showing brain tissue damage [21,56]. Indeed, if lowering the synaptic potential through AMPA receptors delocalization from the synaptic buttons leads to minimizing the glutamate synapse functions, a Hebbian phenomenon would explain the histolopathological result of a lower CA1 thickness in KO mice. The less a synapse is active, the less it is structurally maintained and reversed [57].

## 5. Conclusions

Finally, in the context of acquired or inherited disturbances of methionine synthase activity, our results reveal that synaptic plasticity alterations could contribute to the neurological disorders associated with cognitive impairment. This suggests that the pharmacological modulation of key players in neuronal functions could constitute a promising perspective for developing a novel therapeutic approach aiming at reducing the neurological manifestations, including cognitive impairment, of impaired vitamin B12 metabolism, in the context of genetic diseases or nutritional deficiencies [46]. One strategy could be to promote the phosphorylation of glutamate receptors by targeting kinase pathways putatively linked to other receptors such as estrogen [29] or glucocorticoid receptors [58].

## Figures and Tables

**Figure 1 cells-12-01267-f001:**
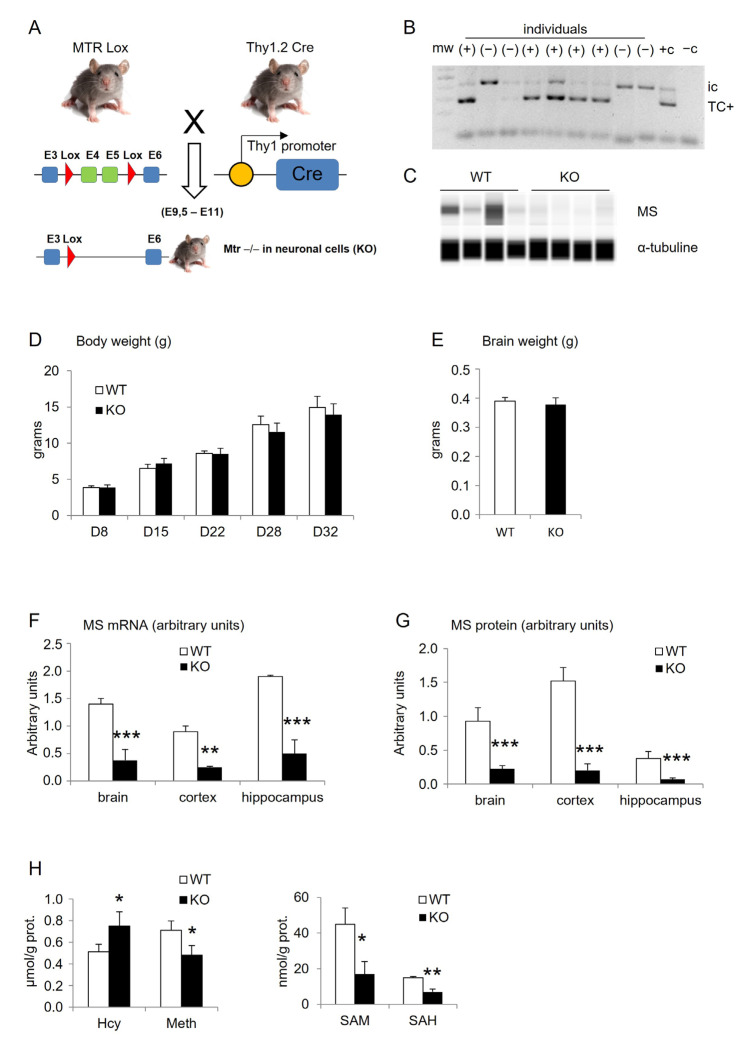
Characterization of mice obtained from the genetic crossing in order to obtain brain-KO-MTR (KO) and controls (WT). (**A**) Schematic representation of the genetic crossing leading to generate offspring depleted in exons 4 and 5 of the *Mtr* gene in neuronal cells. (**B**) Example of the PCR analysis for the cre gene in tail extracts at postnatal day D4 (mw: molecular weight; ic: internal control; TC+: cre gene; +c: positive control; -c: negative control). (**C**) Example of the protein analysis for methionine synthase MS by WES technique at postnatal day D35 for whole brains (MS: methionine synthase). (**D**) Total body weight in grams at different growing time points (D8 to D32, *n* = 9 WT, and 8 KO). (**E**) Total brain weight in gram at D37 (*n* = 9 WT and 8 KO). (**F**) Relative amounts of mRNA for the MTR gene in brain extracts compared to GAPDH (RT-q-PCR, (*n* = 3 WT and 6 KO). (**G**) Quantification of the methionine synthase (MS) protein in brain extracts compared to α-tubuline (WES experiment, *n* = 3 WT and 6 KO). (**H**) Quantifications of markers attesting of the one-carbon metabolism status (SAM, SAH, methyltetrahydrofolate, methylmalonic acid) with the LC-MS/MS technique in hippocampus extracts (*n* = 3 WT and 6 KO). Results are expressed as means ± s.d., ANOVA: * < 0.05; ** < 0.01; *** < 0.001.

**Figure 2 cells-12-01267-f002:**
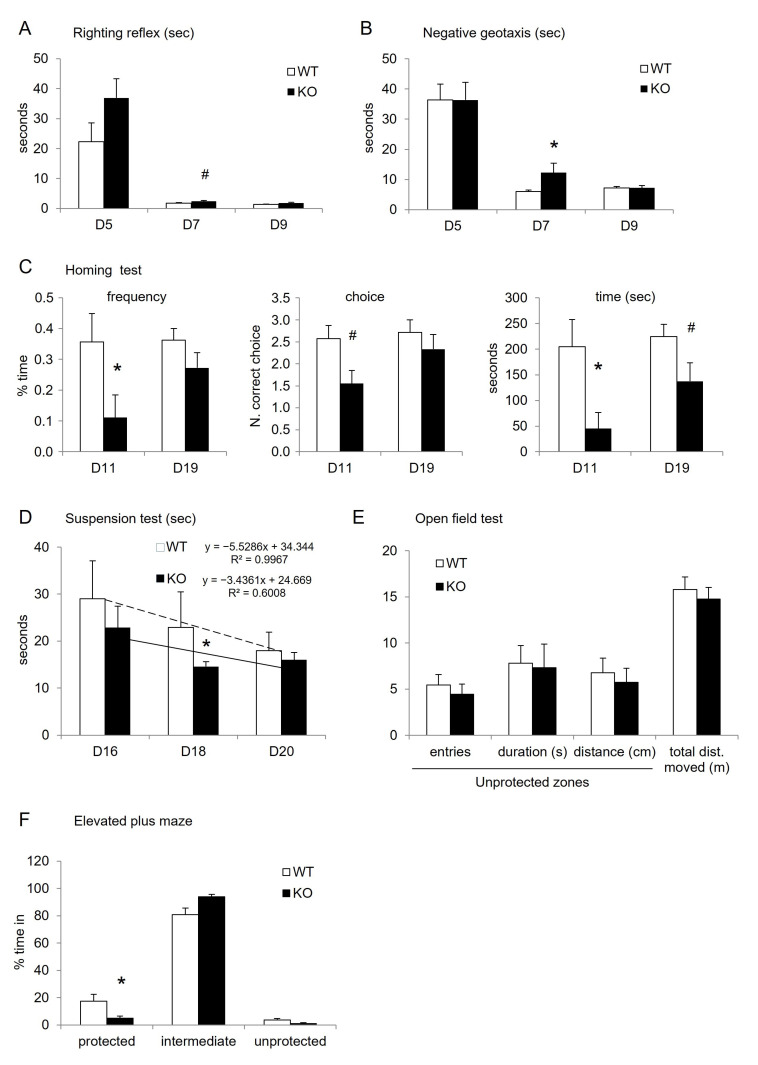
Behavioral characterization of mice (results are expressed as means ± s.e.m., *n* = 8 to 9 WT and 7 to 8 KO; sometimes a mouse failed one test). (**A**) Righting reflex test in seconds at postnatal days D5, D7, D9. (**B**) Negative geotaxis in seconds at postnatal days D5, D7, D9. (**C**) Homing test in frequency to recognize the home odor, amount of first choice for the home odor, and the total time spent close to the home odor in seconds, at postnatal D11 and D19. (**D**) Suspension test performance in seconds before fall at postnatal days D16, D18, and D20 (see statistics below for suspension). (**E**) Open-field test at postnatal day D21. (**F**) Elevated-plus-maze test at postnatal day D22 in the percentage of total time spent in the different zones (F(1,14) = 4.6; *p* = 0.0496). ANOVA: * < 0.05; # tendency < 0.08.

**Figure 3 cells-12-01267-f003:**
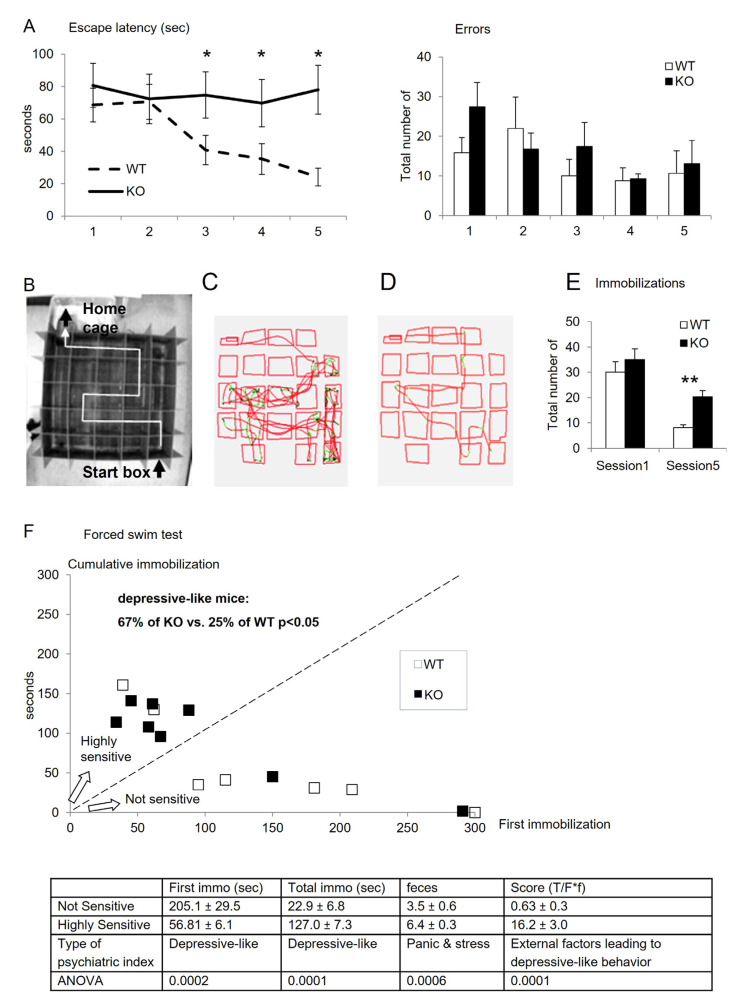
Behavioral characterization of cognitive aspects of mice (results are means ± s.e.m., *n* = 8 to 9 WT and 7 to 8 KO; sometimes a mouse failed one test). (**A**) Aquatic maze performed in 5 successive sessions from postnatal days D28 to D32 with escape latency to find the exit of the maze in seconds and with the total of errors committed (see statistics below for escape latency; comparisons for number of errors were not significant). (**B**) Upper view of the water maze composed of 25 cases, with the ideal route from the start-box to the home-cage (white line). All other cases represent errors and waste of time. (**C**,**D**) Examples of single traces of mice (KO panel C, WT panel D) in the maze at the end of the training sessions (session 5). The red color indicates a rapid speed, while the green + black colors indicate a slow speed and even several stops during navigation, named immobilizations. (**E**) Quantification of immobilizations during the navigation in sessions 1 and 5. (**F**) Porsolt test reveals depressive-like cues at postnatal day D36 in terms of time before the first immobilization in seconds and cumulative immobilizations in seconds. The table summarizes the measurements defining highly sensitive and not sensitive mice in the Porsolt test. ANOVA: * < 0.05; ** < 0.01.

**Figure 4 cells-12-01267-f004:**
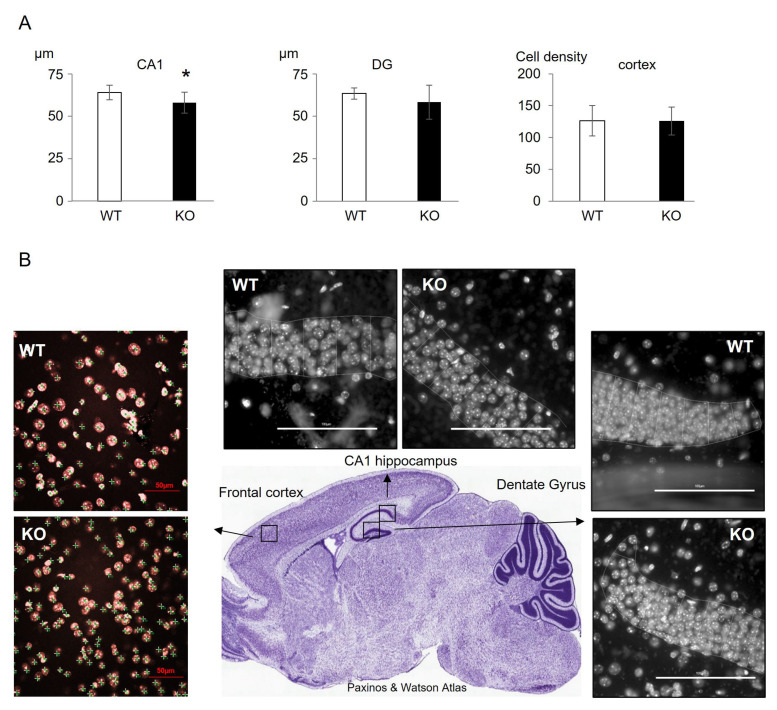
Physiopathology of the hippocampus and frontal cortex. (**A**) Histological quantification of neuronal layers and cell densities in hippocampus (CA1 and Dentate gyrus layers) and frontal cortex layer II. (**B**) Examples of images of the different investigated brain zones, white bars represent 100 µm for hippocampus and red bars represent 50 µm for cortex. Results are means ± s.d., *n* = 4 WT and 4 KO, ANOVA: * < 0.05.

**Figure 5 cells-12-01267-f005:**
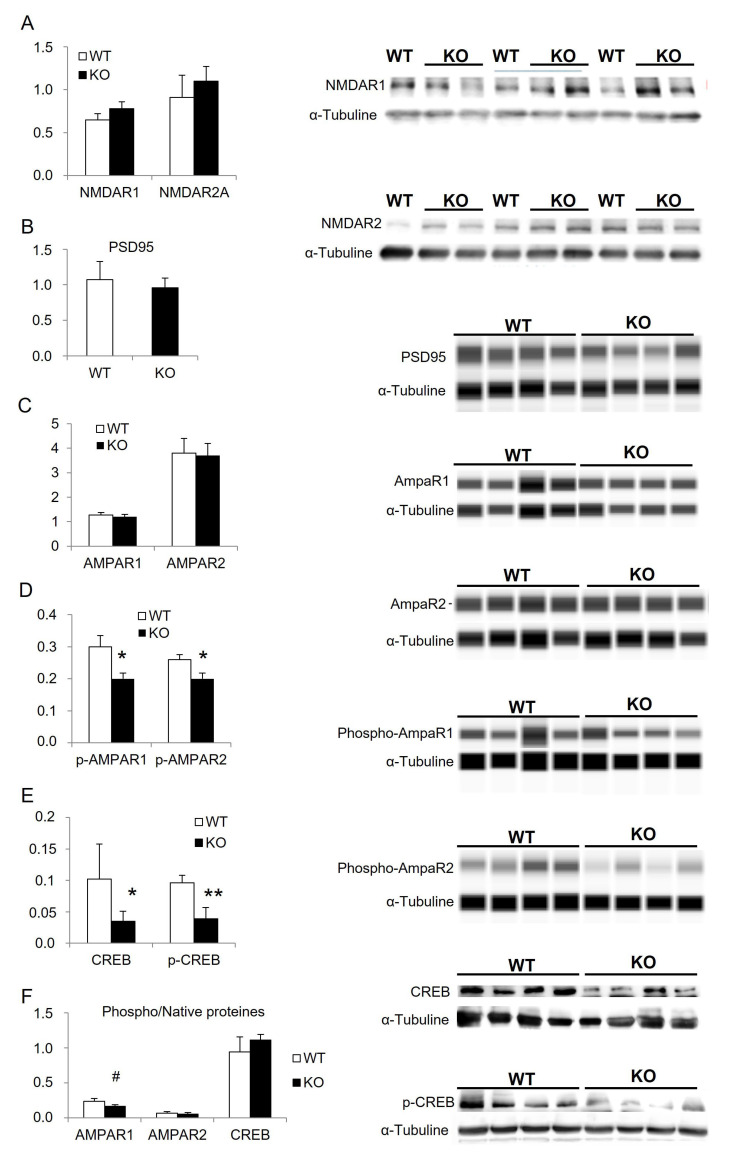
Investigation of glutamate synapses in the hippocampus. Expression of actors of glutamatergic synapses in arbitrary unit standardized with α-tubuline. (**A**) the N-methyl-D-aspartate receptors 1 and 2A; (**B**) the Post-Synaptic-Density-95. (**C**) the amino-propionic-acid receptor 1 and 2. (**D**) phosphorylated amino-propionic-acid receptor 1 and 2. (**E**) transcription factor CREB and phosphorylated CREB. (**F**) ratio phospho/native proteins for AMPAR1, AMPAR2, CREB. Results are expressed as means ± s.d., *n* = 3 to 4 WT and 4 to 5 KO, ANOVA: * < 0.05; ** < 0.01; # < 0.08 tendency.

**Figure 6 cells-12-01267-f006:**
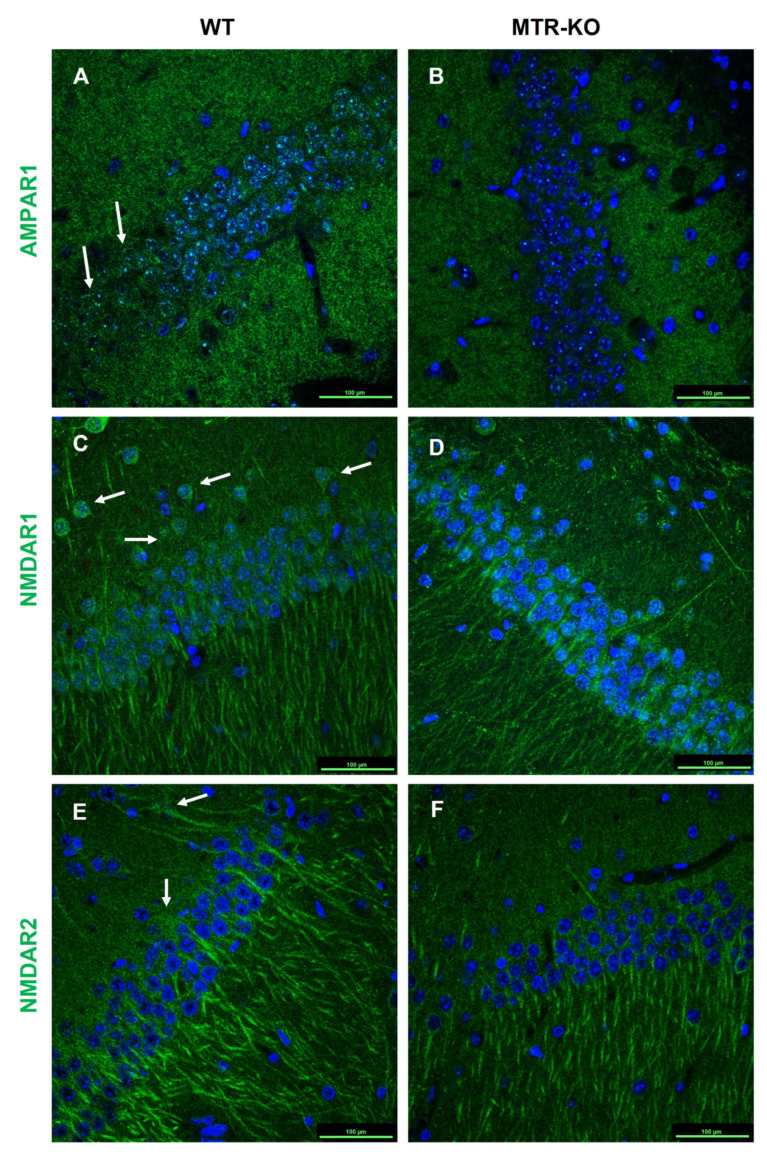
Immunohistochemistry of AMPA and NMDA receptors in the CA1 hippocampus layer at ×60 magnification. Accumulation of green color represents a high density of labeled protein, particularly in WT ((**A**,**C**,**E**), sometimes dots, sometimes fiber-aspect: arrows), compared to KO (**B**,**D**,**F**); bars represent 100 µm (*n* = 4 WT and 4 KO).

**Figure 7 cells-12-01267-f007:**
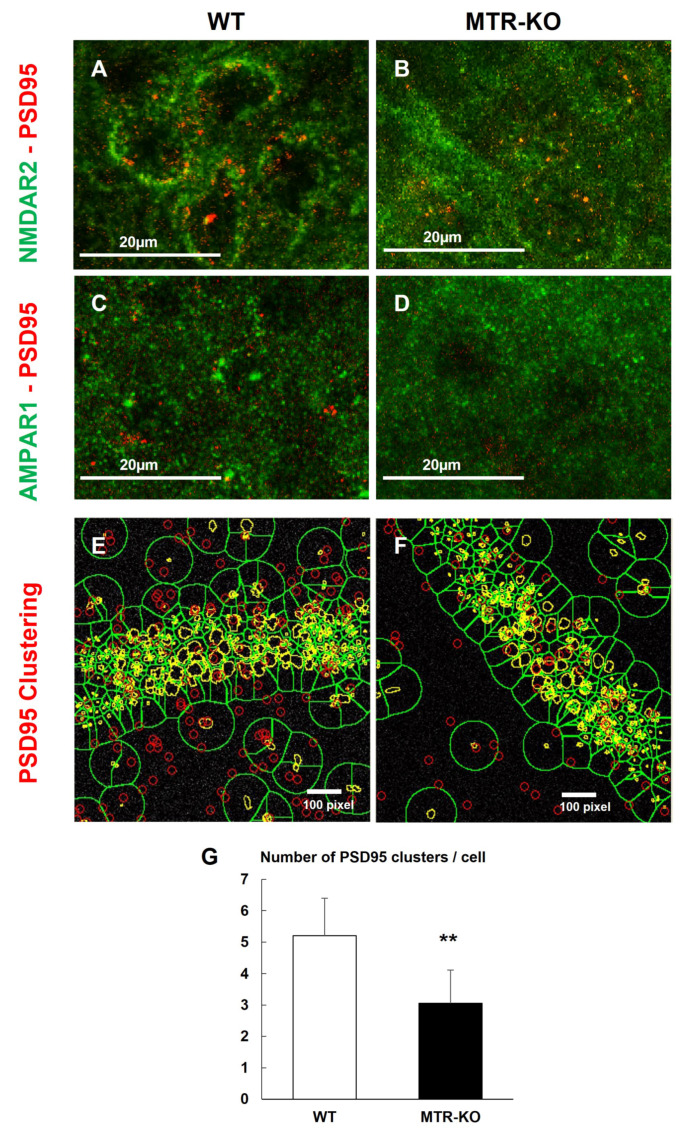
Clustering of AMPA and NMDA receptors with the PSD-95 in the CA1 hippocampus layer at ×100 magnification using the Duo-link technique. See the green accumulation in WT, which appears as a plasma membrane-shaped labeling compared to KO. See the quantity of red dots labeling for protein interactions concerning glutamate receptors with PSD, first (panels **A**–**D**), and secondly, the close accumulation of PSD-PSD, also shown by red dots (panels **E**,**F**); and resulting quantification (**G**). Results are expressed as means ± s.d., *n* = 7 WT and 11 KO, ANOVA: F(1,16) = 15.9, *p* ** = 0.0011.

**Figure 8 cells-12-01267-f008:**
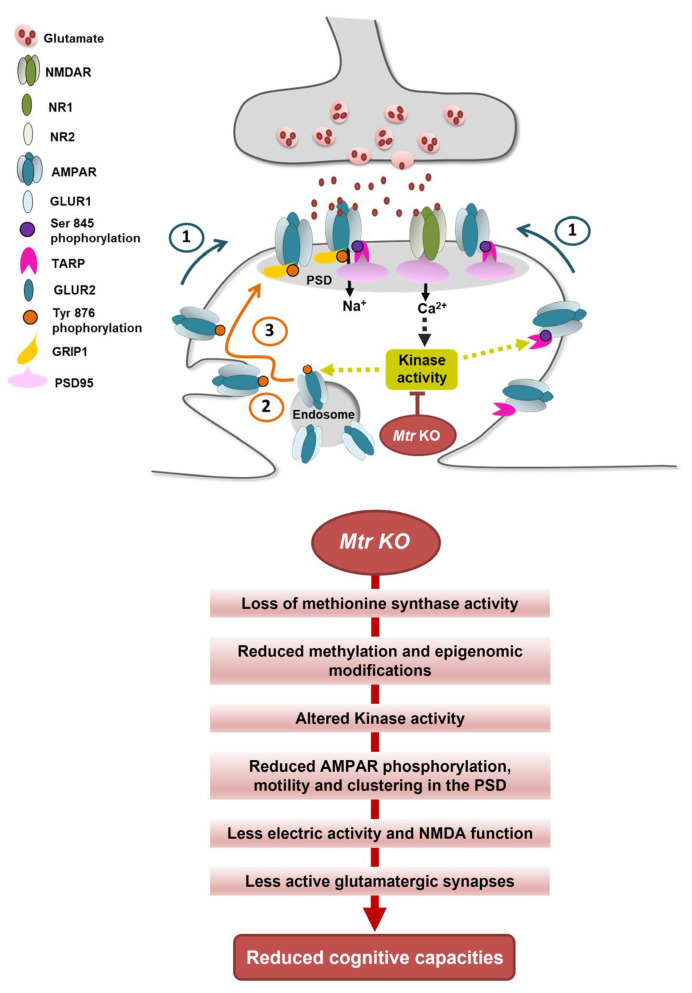
MTR KO alters AMPAR trafficking leading to an alteration of functional plasticity in glutamatergic synapses. The graph presents the mechanisms shown in the present study, our previously published results, and the literature in the field. (1) Pre-existing extra-synaptic surface AMPAR are mobilized to the PSD via lateral diffusion after Tyr 876 and Ser 845 phosphorylation on GLUR2 and GLUR1 subunits, respectively. (2) Tyr 876 phosphoryation of GLUR2 induced AMPAR exocystosis from intracellular endosomes to increase the extrasynaptic surface AMPAR pool, which will migrate laterally to the PSD. (3) AMPAR will be anchored to the PSD and clustered in nanodomain through interaction with Grip1, TARP and PSD95 auxiliary proteins. More AMPA clustering in the PSD favors plasticity and cognitive functions.

**Table 1 cells-12-01267-t001:** Statistics obtained for mRNA, Methionine Synthase proteine and biochemical metabolites, comparisons proposed in Figure 1F–H.

	Brain		Cortex		Hippocampus	
ANOVA qPCR WT vs. KO	F(1,6) = 58.7; *p* = 0.0003		F(1,6) = 28.9; *p* = 0.0017		F(1,6) = 59.2; *p* = 0.0003	
ANOVA MS protein						
WT vs. KO	F(1,6) = 47.6; *p* = 0.0005		F(1,6) = 54.7; *p* = 0.0003		F(1,6) = 44.2; *p* = 0.0006	
ANOVA metabolites	Hcy	Meth		SAM		SAH
WT vs. KO	F(1,6) = 7.0; *p* = 0.0384	F(1,6) = 8.7; *p* = 0.0254		F(1,6) = 10.3; *p* = 0.0184		F(1,6) = 16.0; *p* = 0.0071

**Table 2 cells-12-01267-t002:** Statistics obtained for behavioral tests proposed in Figure 2A–D.

	D5	D7	D9
ANOVA RR test WT vs. KO	F(1,14) = 1.3; *p* = 0.2725	F(1,14) = 9.2; *p* = 0.0090	F(1,14) = 2.8; *p* = 0.1140
ANOVA NG test			
WT vs. KO	F(1,6) = 0.0; *p* = 0.9980	F(1,14) = 4.0; *p* = 0.0645	F(1,6) = 0.06; *p* = 0.8076
ANOVA Homing	D11	D19	
WT vs. KO frequency	F(1,14) = 5.7; *p* = 0.0313	F(1,6) = 0.7; *p* = 0.4168	
WT vs. KO choice	F(1,14) = 4.5; *p* = 0.0529	F(1,6) = 1.9; *p* = 0.1864	
WT vs. KO time	F(1,14) = 7.4; *p* = 0.0165	F(1,6) = 3.6; *p* = 0.0787	
Suspension test two-way ANOVA	ddl	F value	*p* value
genotype	1	4.368	0.0429
days	2	1.716	0.1924
genotype × days	2	1.243	0.2991
Suspension test one-way ANOVA	D16	D18	D20
	F(1,14 ddl)	F(1,13 ddl)	F(1,14 ddl)
WT vs. KO	F = 0.487, *p* = 0.4969	F = 5.389, *p* = 0.037	F = 0.255, *p* = 0.6215

**Table 3 cells-12-01267-t003:** Statistics obtained for the water maze test proposed in Figure 3A.

Water Maze Test								
Two-Way ANOVA		ddl		F Value		*p* Value		
genotype		1		40.887		<0.0001		
days		4		1.925		0.1151		
genotype × days		4		1.669		0.1661		
ANOVA WT sessions								
F(4,40) = 7.436	S2	S3		S4		S5		
S1	ns	0.0037		0.0009		0.0001		
S2		0.0126		0.0034		0.0005		
S3				ns		ns		
S4						ns		
ANOVA F(1,15 ddl)	S1		S2		S3		S4	S5
WT vs. KO	F = 1.978, *p* = 0.180		F = 1.9, *p* = 0.1789		F = 10.8, *p* = 0.0050		F = 13.4, *p* = 0.0023	F = 34.4, *p* < 0.0001

**Table 4 cells-12-01267-t004:** Statistics obtained for protein quantifications in Figure 5.

ANOVA Proteins	*p*-AMPAR1	*p*-AMPAR2	CREB	*p*-CREB
WT vs. KO	F(1,6) = 26.0, *p* = 0.0022	F(1,6) = 9.6, *p* = 0.0213	F(1,6) = 28.9, *p* = 0.0017	F(1,6) = 94.7, *p* < 0.0001

## Data Availability

Not applicable.

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
