# Peer review of "Cognitive Impairment Is Associated with AMPAR Glutamatergic Dysfunction in a Mouse Model of Neuronal Methionine Synthase Deficiency"

_cells, 2023, doi:10.3390/cells12091267_

Round 1

Reviewer 1 Report

While I do not deny that the results are quite interesting, it is not possible to evaluate their solidity and whether the findings support the conclusions because it is often quite difficult to understand what message the authors are trying to communicate. The language needs to be thoroughly revised with the help of a native speaker.

In the abstract, the authors wrote:

“Monitoring the neurobehavioral development of offspring, our results showed that the development did not lead to growth retardation despite an effective reduction of both the expression of the methionine synthase and the methylation status in brain tissues.” What “development” led to growth retardation? I believe that this should read as:

“…………, our results showed that the absence methionine synthase did not lead to growth retardation despite an effective reduction of both its expression and the methylation status in brain tissues.”

The same is valid for the following sentence, which is missing something….

“Several behaviors were differently affected, since only punctual retardations in the acquisition of vegetative functions during the suckling period were recorded AND (?) compared to a dramatic reduction of cognitive performances after weaning. “

The conclusive sentence of the abstract needs better wording …. I do not understand how “the cognitive impairment associated with methionine synthase deficiency could not only result from neurodevelopmental disorders ………” do they mean delay or altered neural development or maldevelopment?

Figure legend to Figure 8 “MTR KO alters AMPAR trafficking and plasticity leading to cognition in glutamatergic synapses.” Not sure what the authors are trying to say.

Beginning of the Discussion: What do the authors want to suggest with (Neurologic retardation?).

What do the authors mean with “neuroretardation”, they should use a more appropriate term.

METHODS

A major concern is the lack of important details and description in the methods, such as a thorough description of the histological analyses, which again has not allowed for a thorough evaluation of the results.

The crossing of the mice should be well described. The first paragraph of the methods should be reorganised in a more consequential manner. Were both male and female mice used for the experiments? “post-developmental time windows” do the authors mean postnatal-development time windows?

At what age were the mice weaned? This seems to be a quite important time-point as many tests were done before or after it but it is not reported………….

At what time of the LD cycle were the behavioural tests performed?

For some of the behavioural tests there is no description of how the different parameters were calculated or what measurements have been done.

What does it mean that “secondary antibodies were used at the provided concentration.” Provided by …..? Maybe the authors mean recommended or suggested by the manufacture?

Paragraph 2.5, were the entire brains used? This is another confusing part, What brain regions were analysed by qPCR and WB, the authors mention ‘microdissection’ but not the brain areas dissected. Did they also use whole brain, including the thalamus, hypothalamus, brain stem etc for some WB and qPCR?

There are no primers reported for Tbp.

Immunohistochemistry: Were the animals perfused?

RESULTS:

The y axes have not labels, these are necessary to fully appreciate the results shown.

How did the authors choose the different days for the behavioural tests? 

Open Field, did the authors measured the time spent in the centre as a measure of anxiety and compared to the time spent in the periphery? Also, what do the authors measure as “entries”?

Same for the elevated plus maze, was the time spent in the centre recorded?

Figure 3C & D are the traces from one mouse or multiple? Is 3D a WT? It would be better if to support their point the authors show the traces from WT and KO mice separately, one mouse/panel will be best so the reader can appreciate the variability and the claim of the authors. 

In the table, Why are the score for the sensitive mice calculated for the WT and KO together? 

The expression “Psychiatric profiles” should be changed, and the following sentence should be explain better “Psychiatric profiles are also related to cognition.” Please add a reference as well.

Figure 4, the authors should provide low magnification images of the WT and KO Hippocampi and frontal cortex to better orient the reader, since the images should have the same orientation and show the same part of the region. Also: What staining are the authors showing? 

There is no description of How the cells were counted, with a software or manually? By who? Did the authors acquire images for the quantifications? How many slices/animal were used? How many fields/slice/animal were used for counting and/or measuring the ‘neuronal layers’? Did they measure the thickness of the cell layers in the CA1, DG and Front Cortex? From where to where? For instance, in the cortex did they begin from the Cajal Retzius layer and measured all the way to the dorsal part of the corpus callosum? How was this performed? What software was used? A thorough description should be provided and maybe an image showing the borders used for the analyses.

Was the scoring of the behavioural assay performed by observers masked to the genotype of the animals? How many observers? Were the histological analyses conducted by masked observers?

It will be best if the authors over-impose the individual samples to the bar graphs to show the animal to animal variability. They should also consult a statistician as some of the data should have been analysed by a two-way ANOVA, as they had two genotype and multiple time points (see Figure 3A and B as an example, but also Figure 2D). How are the data presented? Do the authors show SEM or SD? SD should be used considering the low number of animals.

The number of animals should be reported in the figure legends. In addition, beside the p values and the values from the correlations, all the values generated by the ANOVA and other tests should be provided included the degrees of freedom.

Figure 5E, why the authors are not showing the pCREB/CREB ratio? The actual ‘index’ correlated with cognitive functions is the ratio.

The same is valid for the two forms of the AMPA receptors, usually the ratio between the two forms is also shown and discuss.

Figure 6 and 7, again low magnification for each experimental group should be shown, and a better figure legend describing what the readers are supposed to see should be added.

What is the Duo-link technique?

Figure 7, is the decrease in clustering shown cumulative for both AMPA and NMDA receptors? The authors should show this separately for each receptor. 

The authors should soften some of the claims as they do not have measurements of “neurofunctional deficits”, they can just infer that some might be present due to changes in pCREB protein levels and ratio, some qualitative observations (Fig 6) and decreased number of general PSD95 clusters/cell. 

Author Response

Please find our response in the uploaded file

Reviewer 2 Report

The manuscript ‘Cognitive impairment in a mouse model of neuronal methionine synthase deficiency is associated with AMPAR glutamatergic dysfunction” presents interesting data which are worth publication. The Introduction is well-written and easy to read. However, the rest of the text contains mistakes and parts that need correction. The list of necessary corrections is presented below:

 Line 164 – was the anti-mouse secondary antibody really used? If yes – what for?

Results – please, present the numerical data and p value

Fig. 4 legend – the legend seems to be created for the figure that contains more graphs, the part Fig.4A seems to be missing.

Section 3.3 presents data obtained from experiments that were not mentioned in Methods  - the use of NMDAR1/R2A and PSD-95 antibodies,

Fig. 7 legend – lack of description of individual pictures and a graph (A, B, C … etc.)

Line 391 – “diffused in the hippocampus of KO cells” – rather “diffused in the cells of KO hippocampus”.

Fig. 8 – “glutamate vesicles”  do not exist, change to vesicles containing Glu. “phophorylation” is repeated a couple of times both on the graph and in the legend.

The discussion is slightly too long; moreover, there is no explanation what is the connection between methionine synthase and receptors phosphorylation. What is the direct link? The description of this connection will support the concept presented in Fig. 8. Otherwise the presented concept is only a speculation.

Author Response

(The authors gave the same response as above.)

Round 2

Reviewer 1 Report

The authors have not taken in consideration several of the suggestions provided and have dismissed or not addressed important concerns, others were misinterpreted. Hence, the manuscript needs to be further improved beginning with the language and the grammar.

The methods are still confusing and lacking important details, The figures have not been improved. I did not ask the authors to reduce the length of the discussion, but to soften some of their claims, and I still think that their findings do not support all the claims.

The language has been revised but can still needs to be improved:

Experiments were performed on mice offspring during postnatal growth from 5 days of age (D5) and until after weaning at 37 days of age (D37). “

Please removed “after weaning at” from this sentence as this contradicts what stated  two sentences later, mice were weaned at D21, it just not written clearly….

Neonates were kept with mothers during the lactation period and wean- ing time was considered at D21 after birth.”

In general the first paragraph could be written with better consequentiality:

Mice were maintained under standard laboratory conditions, with a normal 12-hour light/dark cycle, food and water ad libitum. Experimental mice were obtained from males and females C57BL/6 with the Cre/Lox system using Thy-1.2 as a promoter of the Cre recombinase, in order to generate Mtr- knock-out (Mtr-KO) in brain tissue. Wild type animals (WT) were obtained as littermates of KO, when they did not receive and produce the Cre recombinase (figure 1A). Genitor females and males were obtained from ICS (Strasbourg, France). Females carried the "floxed MTR-exons" and males carried the Cre recombinase under the Thy1.2 promotor. Crossing these females and males gave a theoretical range of 50/50% WT and KO off- spring. Neonates were kept with mothers during the lactation period and wean- ing time was considered at D21 after birth. Both males and females offsprings were considered for THE experiments, and when data revealed no sex difference, they were pooled. Experiments were performed on offspring during postnatal growth from 5 days of age (D5) and until after weaning at 37 days of age (D37).

“Both males and females offsprings…..” please remove the ‘s’ from male and female

“mice offspring” please remove ‘mice’ not necessary…

“The left hemisphere were ………………………………… The right hemisphere were …..

Either add an S to hemisphere or change the verb to ‘was’.

“were used according TO manufacturer recommendation ‘

The description in the methods of what brain regions were used in this study and for what assays is still very confusing, can the authors clearly describe when they used Hippocampus and or cerebral cortex.

In this work, as also stated in the rebuttal letter, they mostly or only used hippocampus and cerebral cortex just state that. Microdissections of brain sub-regions does not describe the work, there is no need to report regions/tissue that was collected for other studies, it is just confusing and does not serve this work.

As asked previously: When the whole brain was used what regions were included? Was this the whole right hemisphere or the whole brain?

What does the following sentence mean: “For a global point of view and metabolism analysis, a small piece

of each micro-dissect brains sub-areas was harvested, weighted and mixed (LC- MS/MS quantification). “?

Did they use the entire hippocampus or was it cut in small pieces? If so, what subregions were used? CA1, CA2 or CA 3? How about the dentate gyrus? For the cerebral cortex, what part was used prefrontal? Parietal?

The description of the histological measurements is still lacking the following:

How many sections/animal were used, were these sequential? The authors also report using one slide, but how many sagittal sections were on the slide? Were images acquired from all the sections?

How many images were acquired to measure the thickness of the hippocampal regions? The authors report “that the value for each animal is the mean of 10 measures”, but it is unclear if from one section or multiple.

What software was used?

Were the cells counted on images? How many images were acquired?

Was any software used?

Figure 1C, what tissue/brain area was used ?

Several of the y axis still have no labels………

Figure 4, the reviewer asked that the authors provide a low magnification of the sections from which the images were acquired not a image from an atlas…….

The authors have dismissed several of my comments/suggestions including the one to over-impose the individual samples to the bar graphs to show the animal to animal variability. I really think these should be added, alternatively they could use a violin plot or another graph type that show the sample variability. This representation has now become the norm, not sure why the authors do not want to show these….

Author Response

Authors thank the reviewer for his careful review and all advices. We answer to each points in the following lines.

The authors have not taken in consideration several of the suggestions provided and have
dismissed or not addressed important concerns, others were misinterpreted. Hence, the
manuscript needs to be further improved beginning with the language and the grammar.

We apologize for errors committed in language and grammar. Our colleague Prof. David Meyre from McMaster university, Canada, helped us in editing this last version. We hope to meet a satisfying standard for English language.

The methods are still confusing and lacking important details, The figures have not been
improved. I did not ask the authors to reduce the length of the discussion, but to soften
some of their claims, and I still think that their findings do not support all the claims.

The language has been revised but can still needs to be improved:

The following proposed modifications were applied and corrections were made in the text.

“Experiments were performed on mice offspring during postnatal growth from 5 days of age (D5) and until after weaning at 37 days of age (D37). “
Please removed “after weaning at” from this sentence as this contradicts what stated two sentences later, mice were weaned at D21, it just not written clearly....
“Neonates were kept with mothers during the lactation period and wean- ing time was considered at D21 after birth.”

In general the first paragraph could be written with better consequentiality:
Mice were maintained under standard laboratory conditions, with a normal 12-hour light/dark cycle, food and water ad libitum. Experimental mice were obtained from males and females C57BL/6 with the Cre/Lox system
using Thy-1.2 as a promoter of the Cre recombinase, in order to generate Mtr- knock-out (Mtr-KO) in brain
tissue. Wild type animals (WT) were obtained as littermates of KO, when they did not receive and produce the
Cre recombinase (figure 1A). Genitor females and males were obtained from ICS (Strasbourg, France). Females
carried the "floxed MTR-exons" and males carried the Cre recombinase under the Thy1.2 promotor. Crossing
these females and males gave a theoretical range of 50/50% WT and KO off- spring. Neonates were kept with
mothers during the lactation period and wean- ing time was considered at D21 after birth. Both males and
females offsprings were considered for THE experiments, and when data revealed no sex difference, they were
pooled. Experiments were performed on offspring during postnatal growth from 5 days of age (D5) and until
after weaning at 37 days of age (D37).
“Both males and females offsprings.....” please remove the ‘s’ from male and female
“mice offspring” please remove ‘mice’ not necessary...
“The left hemisphere were ....................................... The right hemisphere were .....
Either add an S to hemisphere or change the verb to ‘was’.
“were used according TO manufacturer recommendation ‘

The description in the methods of what brain regions were used in this study and for what assays is still very confusing, can the authors clearly describe when they used Hippocampus and or cerebral cortex.
In this work, as also stated in the rebuttal letter, they mostly or only used hippocampus and cerebral cortex just state that. Microdissections of brain sub-regions does not describe the work, there is no need to report
regions/tissue that was collected for other studies, it is just confusing and does not serve this work. As asked previously: When the whole brain was used what regions were included? Was this the whole right hemisphere or the whole brain?

Precisions are given in the section 2.1. In the beginning of the study, whole brain were used for total weight, MTR mRNA and MS protein; and after these global investigations, hippocampus and cerebral cortex were microdissected from the right hemisphere.

What does the following sentence mean: “For a global point of view and metabolism analysis, a small piece of each micro-dissect brains sub-areas was harvested, weighted and mixed (LC- MS/MS quantification). “?

Did they use the entire hippocampus or was it cut in small pieces? If so, what subregions were used? CA1, CA2 or CA 3? How about the dentate gyrus? For the cerebral cortex, what part was used prefrontal? Parietal?

We modified the section 2.5. of the methods. Since the LC-MS/MS is a high precise technique, a very small amount of tissue is sufficient. Thus, a piece of hippocampus was taken. The structure of the hippocampus was conserved and cut transversally for LC-MS/MS analysis. Indeed, all hippocampus layers were used; CA1, CA2, CA3, Dentate gyrus.

For the cerebral cortex, the structure was also conserved during dissection, and cut in two according parietal zone. The frontal piece were used for our analysis.

The description of the histological measurements is still lacking the following:
How many sections/animal were used, were these sequential? The authors also report using one slide, but how many sagittal sections were on the slide? Were images acquired from all the sections?
How many images were acquired to measure the thickness of the hippocampal regions? The authors report
“that the value for each animal is the mean of 10 measures”, but it is unclear if from one section or multiple.

2 slides per individual considered with the same atlas coordinate at ±100 µm

What software was used?

Olympus CellF software

Were the cells counted on images?

All cells on a 500x500µm square

How many images were acquired?

2 images (1 on each slide according the atlas coordinate)

Was any software used?

Olympus CellF software

Figure 1C, what tissue/brain area was used ? whole brains (see figure 1 legend)
Several of the y axis still have no labels.........All have, except figure 2E, because of multiple unities known with x axis

Figure 4, the reviewer asked that the authors provide a low magnification of the sections from which the images were acquired not a image from an atlas.......

Several weeks will be necessary to perform a new experiment in order to obtain new slides for an image with very low magnification. We do not conserve such very low magnification allowing to have a whole mouse brain in a single view. Ask if absolutely necessary.

The authors have dismissed several of my comments/suggestions including the one to over-impose the individual samples to the bar graphs to show the animal to animal variability. I really think these should be added, alternatively they could use a violin plot or another graph type that show the sample variability. This representation has now become the norm, not sure why the authors do not want to show these...

We do not understand this remark. May we ask if the reveiwer had the last version of the revised figures? We add bar graphs on each presented images. To do so, we returned to each original image conserved in the microscope computer and standardized each bar.

Concerning the remark "Note also, that the conclusions should be soften as they are not fully supported by the results", we propose a new version of the second half of the discussion as followed:

Last version :Nevertheless, the phosphorylation of AMPA receptors was significantly reduced in KO mice, suggesting a deregulation of the functional synaptic plasticity.

We agree that investigating the synaptic plasticity would be directly shown by electrophysiology, for example. The sentence is modified for less speculation:

Nevertheless, the phosphorylation of AMPA receptors was significantly reduced in KO mice, suggesting a deregulation of AMPAR functionality. This proposed mechanism is reinforced by the fact that the measurements concerning glutamate synapses in the hippocampus were performed on tissue extracts collected only three days after the maze test and immediately after the forced-swim test

Last version : In our neuronal-specific model of methionine cycle deregulation, it appears that neurofunctional deficits, such as cognitive dysfunctions also monitored in patients, are at least related to a loss of AMPAR electric activity. Such first reduced activity is known to drive NMDAR-related plasticity leading to cognitive outcomes after brain formation in early age. In addition, this mechanistic proposal could also explain the slight decrease we obtained in the CA1 hippocampus thickness in KO mice, contrary to most nutritional models showing brain tissue damage [22,59].

We agree that our study did not present the electric activity and was focused on AMPA receptors and not on NMDA ones, so we discuss now only in the light of AMPAR.

We propose a less affirmative, less speculative sentence:

In our neuronal-specific model of methionine cycle deregulation, it appears that neurofunctional deficits, such as cognitive dysfunctions also monitored in patients, correlate with a loss of AMPAR recruitment to the membrane. Such an AMPAR-mediated reduced synaptic activity is known to drive NMDAR-related plasticity leading to cognitive functions. In addition, this mechanistic link starting with AMPAR could also explain the slight decrease we obtained in the CA1 hippocampus thickness of KO mice, contrary to most nutritional models showing developmental brain damages [22,59].

Between these two paragraphs, we linked results of the present study with other studies to agree that the clustering and addressage of AMPA receptors to the Psot Synaptic Density (PSD) was clearly presented as a high relevant mechanism linked to plastic functionality. Another point that reviewers apparently did not know is the Duo-Link technique that allow such clustering measurements. So, we agree to be less speculative in particular points of the manuscript, but we discuss our results in the light of related literature, which is scientifically usual.

Reviewer 2 Report

There was a small misunderstanding; writing that authors should include numerical data, I meant presenting numerical data in places where it was said that something increased or decreased.  (For example “it appeared that KO mice need more stops and immobilizations, especially at corners of the ideal route (figure 3C, green+black colors) compared to WT” – how many (or what %) stops less?)

However, It is great that the authors decided to show numerical data in tables, but these tables should be treated as separate creations, and each should have a specific title and references in the text (Table 1, etc.)

Surprisingly, the presented revised manuscript does not include figures, which is a bit surprising and disappointing. Especially since there were a few errors in Fig.8 which correction should be possible to check.

Page 12, line 498 – please, change “phophorylation” to phosphorylation.

Author Response

Remarks and responses round 2

There was a small misunderstanding; writing that authors should include numerical data, I meant presenting numerical data in places where it was said that something increased or decreased.  (For example “it appeared that KO mice need more stops and immobilizations, especially at corners of the ideal route (figure 3C, green+black colors) compared to WT” – how many (or what %) stops less?)

Yes, a misunderstanding of author to your remark. We were focused to the replay concerning statistics and we made a confusion. We now propose several numerical data in the text of the Result section.

However, It is great that the authors decided to show numerical data in tables, but these tables should be treated as separate creations, and each should have a specific title and references in the text (Table 1, etc.)

We now propose separated tables with titles. But We also propose to transfer them in supplementary information. Such a choice could be validated by the editor.

Surprisingly, the presented revised manuscript does not include figures, which is a bit surprising and disappointing. Especially since there were a few errors in Fig.8 which correction should be possible to check.

Sorry, but we up-loaded the modified text without figure because it was too heavy. The figure were corrected in the "figure file"

Page 12, line 498 – please, change “phophorylation” to phosphorylation.

Sorry. It is corrected